# The Effects of Lutein-Containing Supplement Intake on Glycation Inhibition Among Diabetic Patients with Cataracts

**DOI:** 10.3390/ijms26125706

**Published:** 2025-06-13

**Authors:** Rijo Hayashi, Shimmin Hayashi, Shigeki Machida

**Affiliations:** 1Department of Ophthalmology, Saitama Medical Center, Dokkyo Medical University, Koshigaya 343-8555, Saitama, Japan; livelyeye@mopera.net (S.H.); machidas@dokkyomed.ac.jp (S.M.); 2Lively Eye Clinic, Soka 340-0053, Saitama, Japan

**Keywords:** antioxidant, lutein, diabetes, senile cataract, glycation, advanced glycation end products, aqueous humor

## Abstract

Glycation is known as an important factor inducing human diseases, including diabetic complications. As oxidative stress contributes to procedures of glycation, antioxidants may inhibit glycation and delay the progression of diabetic complications. Our previous investigation of human aqueous humor after the intake of a lutein-containing supplement demonstrated increases in antioxidative activities and decreases in peroxidative products. This study enrolled 25 patients with diabetes (DM group) and 100 age-matched controls. Aqueous humor samples were collected during cataract surgery before and after 6 weeks of oral intake of the lutein-containing antioxidant supplement, Ocuvite + Lutein^®^. The carboxymethyl-lysine level (CML) was measured as an indicator of glycation. Levels of superoxide dismutase activities (SOD) and total hydroperoxide (TH) were measured as indicators of oxidation. Changes after intake and the differences between age-matched controls and the DM group were evaluated. CML decreased after intake among the DM group, while there were no changes among the age-matched controls. SOD was significantly lower and TH was significantly higher in the DM group as compared to the age-matched controls, both before and after intake. In line with the decreases in glycation, the intake of lutein-containing antioxidant supplements may inhibit diabetic complications in diabetic patients.

## 1. Introduction

Age-related cataracts (ARC) play a significant role in the increasing level of avoidable vision impairment, second to uncorrected refractive error [1]. Although ARC can be surgically treated, in many developing nations, this is cost-prohibitive and makes up a large portion of medical expenses. Nearly 30% of ophthalmological medical expenses have been reported to be related to the treatment of ARC among Japanese patients over 65 years of age [2].

Oxidative stress is known as one of the significant factors that can induce ARC [3,4,5,6,7]. Previous studies have evaluated the correlations between decreases in antioxidant substances, such as superoxide dismutase (SOD), glutathione, and ascorbic acid, and the increasing incidence and progression of ARC [3,4,5]. Protein glycation, followed by crosslinking that subsequently results in aggregation [8,9,10,11], has also been suggested to be a risk factor of ARC [12]. To initiate protein glycation, a carbonyl group of reducing sugars will conjugate with free lysine or arginine of proteins [13]. The formation and rearrangement of a reversible Schiff base will generate Amadori products [14], which can then undergo several reactions to form more stable advanced glycation end products (AGEs). AGEs have been observed during the normal aging process, while higher serum levels have also been reported in diabetes mellitus (DM) [15,16]. Numerous AGEs have been discovered in the human lens [17,18], including carboxymethyl-lysine (CML). Higher serum levels of AGEs have been reported in ARC patients, both in diabetic and non-diabetic patients [19,20,21,22]. AGEs have also been considered to play a significant role in ARC formation.

Many reviews have investigated nutrients and their effect on eye health [23,24,25,26,27,28]. A meta-analysis of cohort studies has concluded that a higher consumption of nutrients, such as vitamins A, C, and E, lutein, and zeaxanthin, was associated with a reduced risk of ARC [29]. The results of our previous study, which investigated the same person with ARC before and after consuming a lutein-containing antioxidant supplement, demonstrated that there were increases in the levels of the superoxide scavenging activities and decreases in the levels of total hydroperoxides (TH) in the aqueous humor [30]. As previously described, both protein glycation and oxidative stress are thought to play significant roles in cataract development. Antioxidants have been reported to be promising in inhibiting the progress of cataracts by decreasing oxidative stress. The aim of this study was to investigate whether antioxidant nutrients also inhibit glycation in diabetic patients. This is the first investigation measuring the changes of AGEs in aqueous humor of the same patient with cataracts and DM before and after the intake of an antioxidant supplement.

## 2. Results

There were no significant differences in the age between the DM and age-matched controls, in both genders (Table 1). Among males in the DM group, the CML levels in the post-intake samples were significantly lower compared to those for the pre-intake samples (Figure 1a). The same tendency was observed among the females in the DM group, although it did not reach statistically significant levels. There were no significant differences between the genders in either the pre-intake or post-intake samples. There was a significant correlation between the CML levels in the pre-intake samples and those in the post-intake samples among males (Figure 1b). There were no significant differences between the SOD in pre-intake and the post-intake samples, for either males or females (Figure 2a). There was no correlation between the SOD activities in the pre-intake samples and those observed in the post-intake samples among both genders (Figure 2b). There were no significant differences between the TH levels in pre-intake and post-intake samples, for either males or females (Figure 3a). There was also no correlation between the TH levels in pre-intake samples and those in the post-intake samples among both genders (Figure 3b).

In the age-matched controls, there were no significant differences between the CML levels in pre-intake and post-intake samples, for either males or females (Figure 4a). There was a significant correlation between the CML levels in the pre-intake samples and those in the post-intake samples among both genders (Figure 4b). There were no significant differences between the genders in either the pre-intake or post-intake samples.

With regard to the comparison between the DM and age-matched controls, there were no significant differences in the CML levels, in either the pre-intake or post-intake samples in both genders (Figure 5). However, SOD activities were significantly lower in the DM group among both genders, for both the pre-intake and post-intake samples (Figure 6). In addition, TH levels were significantly higher among males for both the pre-intake and post-intake samples, while this was only observed in post-intake samples among the females (Figure 7).

## 3. Discussion

After 6 weeks of lutein-containing antioxidant supplement intake, the CML levels in the aqueous humor decreased in the DM group, which was statistically significant in males and with a decreasing tendency in females. In contrast, there was no increase in the SOD activities or any decrease in the TH among the DM group. In contrast, there were no changes in the CML levels, while there were increased SOD activities and decreased TH in the age-matched controls. These findings suggest the possibility that lutein-containing supplements have different effects in patients with DM as compared to those without DM. After lutein-containing supplement intake, there was a decrease in the glycation in DM patients, while there was an increase in the antioxidative activities in the non-DM patients.

Oxidative stress and glycation are both associated with the aging process [31,32]. Higher levels of AGEs, such as CML, as well as increased oxidative stress, were reported in the healthy aged population [33]. AGE-related changes during aging are associated with the generation of reactive oxygen species (ROS) [33,34,35,36], which further induce the formation of AGEs [37]. The interaction of AGEs with the receptor of AGEs (RAGE) also activates NAPDH oxidases, thereby enhancing the intracellular ROS generation [34,35]. Other studies have also suggested that AGEs can additionally stimulate ROS generation through the mitochondrial electron transport chain [36]. ROS enhancement, in turn, contributes to AGE formation [37] and RAGE expression [38]. This creates a vicious reaction cycle, which progressively increases ROS and AGEs. In the lens, both protein glycation [12] and oxidative stress [6,7] are thought to play significant roles in ARC development. It is possible that increasing antioxidative capacities followed by glycation inhibition plays a key role in inhibiting the development of the aging process. Our previous investigation of the aqueous humor and anterior capsules collected from cataract surgery indicated that there were increases in the antioxidative capacities and decreases in the oxidative products after lutein-containing antioxidant supplement intake [30,39,40]. The results of this study indicate that AGEs also decreased after supplement intake in DM patients. Furthermore, these findings additionally suggest the possibility that the intake of antioxidants, including lutein, will induce increases in antioxidation and suppression in glycation followed by inhibiting the formation and progression of cataracts in DM patients. On the other hand, in non-DM patients, AGEs did not change after intake of the lutein-containing antioxidant supplement. It is possible that glycation is not as severe as in DM patients and, thus, the increased antioxidative capacities will not change glycation.

The results of this study revealed that there were no differences in the AGEs of aqueous humor between DM and non-DM patients, which is in line with the report by Franke et al. [41]. Study results reported by Chitra et al. indicated that there was a higher serum AGE index in cataract patients as compared to non-cataract patients, which was not different from the difference found between the DM and non-DM group [42]. In another study, a significantly high serum AGE index was only reported in patients with diabetic complications, such as retinopathy, and was not observed in diabetic patients without complications [43]. AGEs in vitreous [44] and from the skin [45,46] were reported to be positively correlated with blood HbA1c levels. These reports support the results of this study, which found AGEs in the aqueous humor of DM patients without diabetic complications, which did not differ from what was observed in non-DM patients. Furthermore, as only well-controlled DM patients were enrolled in this study, it is possible that the capacities of handling ROS were reduced and barely kept the AGEs within normal levels. It is very possible that AGEs would increase if the DM in these patients became poorly controlled and the hyperglycemia progressed.

Although there were no differences in AGEs levels, SOD was lower and TH was higher in the DM versus the non-DM patients in this study. It is possible that capacities in handling ROS could increase in DM patients after the intake of antioxidants as compared to non-DM patients. However, these increases in the capacities for handling ROS involve the inhibition of the formation of glycation products, and thus, there are no obvious increases in the SOD. This may be the reason why the AGEs decreased but SOD did not increase, or why TH did not decrease in the DM patients after the antioxidant intake in this study.

In addition to the endogenous pathway products, AGEs can also accumulate from dietary sources. Furthermore, enrolled patients might be aware that antioxidant intake can be effective healthwise and thus, this may have led to the patients converting to low AGEs and high antioxidant diets. As all of the patients enrolled in our study were instructed not to change their diets, the decreases in the AGEs should not have been due to changes in diet. The results of this study that showed CML levels in the pre-intake samples were positively correlated with those in the post-intake samples in both DM and non-DM patients, possibly due to the unchanged diet. However, in an investigation that examined aged mice [47], the study results showed that a low AGE diet induced decreases in the ROS and AGEs generated in the body. Furthermore, RAGE was also suppressed. In contrast, under conditions of chronically excessive exogenous AGEs, the capacity for handling ROS was exceeded. Thus, in addition to antioxidant supplement intake, low AGEs and high antioxidant diets are strongly recommended.

Although the changes after the intake of this lutein-containing supplement in DM patients exhibited the same tendencies among both genders, the results of our previous investigations in non-DM aged patients suggested that the intake of antioxidants exhibits effects that differ between the genders. After lutein-containing supplement intake, SOD increased and TH decreased in the aqueous humor of both genders [30]. However, H_2_O_2_ decreased only in males. This suggests that SOD increases in both genders, while the H_2_O_2_ scavenging activity in females does not increase proportionally as it does in males. Gene expression in lens epithelial cells also changes in different ways between the genders. The expression levels of Glucose-6-phosphate dehydrogenase and 18S RNA were found to be significantly higher among female patients [39], while the aquaporin 8 expressions were lower among males [40]. After the intake of these lutein-containing antioxidants, there was also an increase in the macular pigment optical density, which was greater in females compared to males [48]. Based on these results, it is highly probable that the intake of antioxidants is effective at suppressing oxidation through different mechanisms that occur between the genders. Estrogens are known to have antioxidant effects [49,50] due to the direct scavenging of free radicals [51] and the upregulation of antioxidative enzymes [52]. In post-menopausal female patients, peroxidation increases due to the dramatically decreased levels of estrogen. Although the differences in these mechanisms among both genders remain unclear, these are likely associated with the effects of estrogen. Gender-differentiated analyses are strongly recommended when interpreting the effects of antioxidant supplementation on age-related diseases. The results of this study demonstrate that there were no significant differences in the antioxidation and anti-glycation effects for this lutein-containing antioxidant supplement between the genders. Thus, it is possible that the gender differences in DM patients differ from those found in non-DM patients. It is strongly recommended that further investigations between DM and non-DM patients be conducted.

The limitations of this study included the fact that only patients with cortical cataracts were evaluated, which are considered to be age-related. Thus, evaluations of patients with posterior cataract, which is known to be one of the diabetic complications, are recommended. In addition, as described above, the AGEs in the aqueous humor of DM patients without diabetic complications did not differ from those observed in non-DM patients. Thus, the levels of AGEs in the aqueous humor among DM patients with diabetic complications could possibly differ from those among the well-controlled DM patients without diabetic complications. Moreover, the effects of this lutein-containing supplement could possibly differ among poorly controlled DM patients that do have diabetic complications. Therefore, it is strongly recommended that further investigations that evaluate poorly controlled MD patients be conducted.

The supplement utilized in this study, Ocuvite + Lutein^®^, includes lutein, Vit C, Vit E, and β-carotene. Vit C is known as an effective water-soluble antioxidant [53,54] and Vit E is known as a lipid-soluble antioxidant that inhibits the peroxidation of lipids [55]. Vit C is known to have synergistic effects with Vit E in protecting the cell membrane from oxidative damage [56,57]. Synergistic effects of Vit E, β-carotene [58], and multivitamins [59,60] were reported. The effects of the Ocuvite + Lutein^®^ on the mechanisms of increased superoxide scavenging activity and decreased oxidation are possibly due to the cooperation of lutein with vitamins included in the supplement.

Decreases in AGEs after lutein-containing antioxidant intake in DM patients might not be limited to just the aqueous humor. Our previous investigation reported finding increases in the macular pigment optical density [48] and lutein levels in the aqueous humor and serum [61] after lutein-containing antioxidant supplement intake. These findings demonstrated that this lutein-containing antioxidant supplement might also increase the antioxidative capabilities and decrease the AGE formation in the serum and vitreous, in addition to inhibiting diabetic ocular complications, such as retinopathy. Furthermore, it is strongly recommended that investigations on the possibility of being able to inhibit systemic diabetic complications be conducted.

## 4. Materials and Methods

Participant recruitment was essentially the same as that described in detail in our previous study [39]. Briefly, this study enrolled patients who had Grade 3 to Grade 4 cortical cataract according to the Lens Opacities Classification System III [62], and who had the same grade of lens opacity in both eyes. Patients with ocular complications (such as uveitis or retinopathy), systemic disease except diabetes, or those taking other supplements were excluded. Diabetic patients with poor control, which was defined as having an HbA1c over 7.5%, were also excluded. A total of 125 patients were enrolled in our analyses, which included 25 patients with diabetes (10 males and 15 females, DM group) and 100 patients without diabetes (45 males and 55 females, age-matched controls). All patients provided informed consent, and the study followed the tenets of the Declaration of Helsinki. Approval was granted by the Institutional Human Experimentation Committee of the Saitama Medical Center, Dokkyo Medical University (Approval number: 2069).

The treatment protocol used in this study was essentially the same as that described in detail in our previous study [24]. The aqueous humor was collected during the surgery (as the pre-intake samples). After 6 weeks of Ocuvite + Lutein^®^ intake, samples of the aqueous humor were collected during the surgery for the other eye (as the post-intake samples). Table 2 presents the composition of the Ocuvite + Lutein^®^ supplement. All enrolled patients were instructed not to change their daily diet during the intake of the supplement.

Aqueous humor samples were filled with nitrogen gas and stored at −40 °C until measured. As an indicator of glycation, concentrations of CML were measured using the Enzyme-Linked Immunosorbent Assay Kit for CML (Cloud-Clone Corp, Katy, TX, USA). Aqueous humor samples were added to a microplate, which was pre-coated with a monoclonal antibody specific to CML. Biotin-labeled CML was added to initiate a competitive inhibition reaction with CML in the aqueous humor sample, which was unlabeled, and then incubated for 1 h at 37 °C. After washing off unbound conjugate, avidin conjugated to horseradish peroxidase was added to the conjugate with biotin, which was labeled with CML and incubated for 30 min at 37 °C. After washing out the unconjugated material, TMB substrate solution was added and incubated for 15 min at 37 °C, followed by adding 0.16 M sulfuric acid as the stop solution. Subsequently, the intensity of color that developed was measured at 450 nm. The color intensity amount, which represented the concentration of horseradish peroxidase conjugated CML, was inversely proportional to that of the CML in the sample.

SOD, which was used as an indicator of the antioxidation activities in the aqueous humor, was measured using OxiSelect^TM^ Superoxide Dismutase Activity Assay Kit (Cosmo Bio Co., Tokyo, Japan). Aqueous humor samples were added to a microplate with xanthine solution, while the chromogen solution was added to the xanthine oxidase solution to generate superoxide. After incubation for 1 h at 37 °C, the intensity of the color that developed was measured at 490 nm. The color intensity amount, which represented the concentration of superoxide, was inversely proportional to SOD activities in the sample. The levels of TH were used as the indicators of oxidative stress, and included hydrogen peroxide and the peroxides of lipids, peptides, proteins, nucleic acids, and nucleotides. TH levels were measured using a method modified from the derivatives of reactive oxidative metabolites (d-ROMs) test. The details are described in our previous report [24].

Comparisons between the pre- and post-intake samples were analyzed using a paired *t*-test, while comparisons between genders were analyzed with a *t*-test. All of the correlations were analyzed with regression analysis. Statistical significance was set at a *p*-value of 0.05. All the statistical analyses were performed with IBM^®^SPSS Statistics 28.

## 5. Conclusions

The findings of this study demonstrated that the intake of antioxidants, including lutein, may inhibit glycation in diabetes, especially in males. These results suggest the possibilities of inhibiting cataract progression and diabetic complications in this patient group.

## Figures and Tables

**Figure 1 ijms-26-05706-f001:**
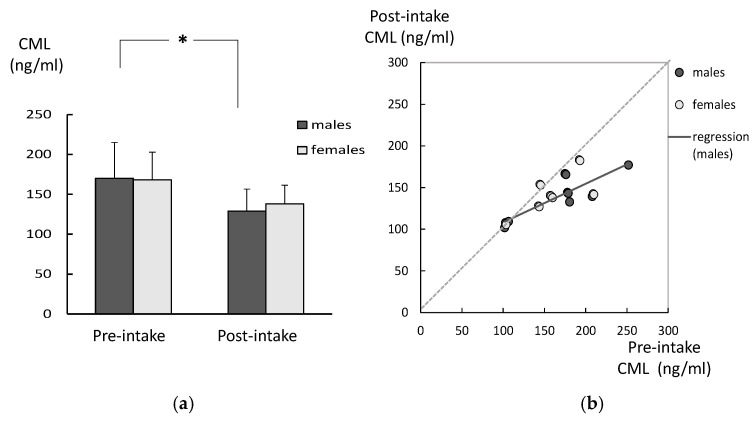
Changes in CML levels of the DM group after 6 weeks of lutein-containing supplement intake. (**a**) Among males in the DM group, CML levels in the post-intake samples were significantly lower compared to those observed in the pre-intake samples (*p* = 0.0192, analyzed with paired *t*-test). Although the same tendency was found among females in the DM group, this did not reach statistically significant levels. There were no significant differences between the genders for either the pre-intake or post-intake samples (analyzed with *t*-test). (**b**) There was a significant correlation between the CML levels in the pre-intake samples and those in the post-intake samples among males (analyzed with regression analysis). Although the same correlation was not shown among females, the CML level values for females are located on the right portion of the graph below the regression line, which indicates decreases in the CML levels after the intake. * *p* = 0.0192.

**Figure 2 ijms-26-05706-f002:**
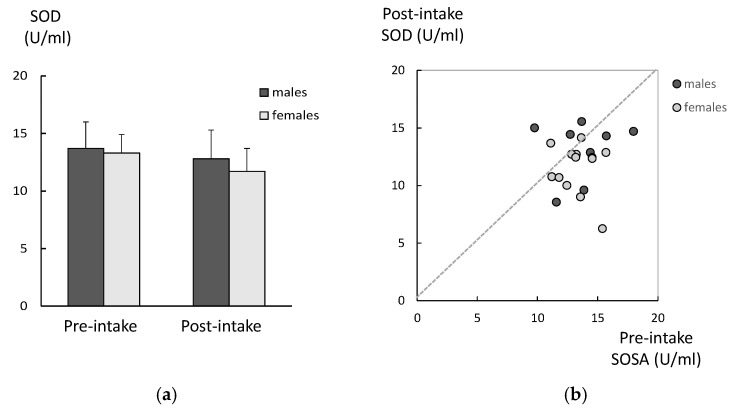
Changes in SOD levels of the DM group after 6 weeks of lutein-containing supplement intake. (**a**) There were no significant differences between the SOD levels in the pre-intake and post-intake samples (analyzed with paired *t*-test) in either males or females (analyzed with *t*-test). (**b**) There was no correlation between the SOD levels in the pre-intake samples and the post-intake samples among both genders (analyzed with regression analysis).

**Figure 3 ijms-26-05706-f003:**
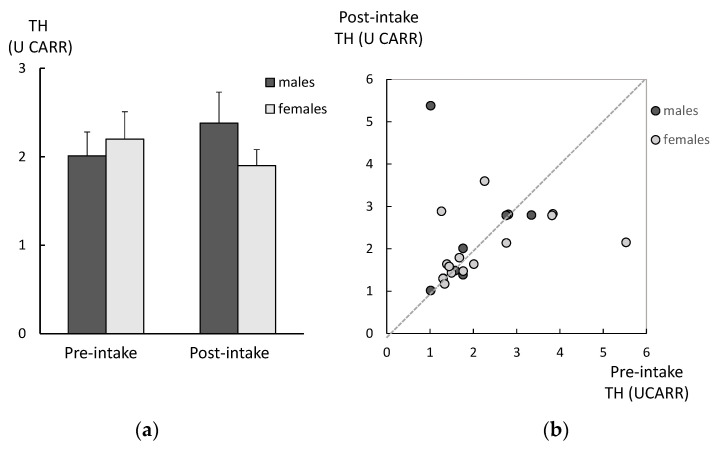
Changes in TH levels of the DM group after 6 weeks of lutein-containing supplement intake. (**a**) There were no significant differences between the TH levels in the pre-intake and post-intake samples (analyzed with paired *t*-test) in either males or females (analyzed with *t*-test). (**b**) There was no correlation between the TH levels in the pre-intake samples and post-intake samples among both genders (analyzed with regression analysis).

**Figure 4 ijms-26-05706-f004:**
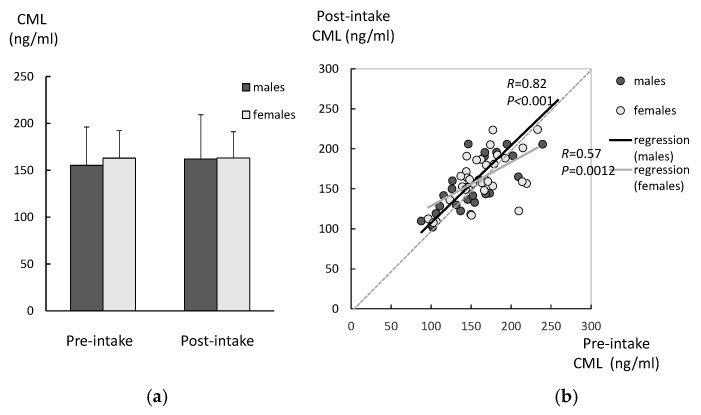
Changes in the CML levels of the age-matched controls after 6 weeks of lutein-containing supplement intake. (**a**) In the age-matched controls, there were no significant differences between the CML levels in the pre-intake and post-intake samples (analyzed with paired *t*-test) in either males or females. (**b**) There was a significant correlation between the CML levels in the pre-intake samples and post-intake samples among both genders (analyzed with regression analysis). There were no significant differences between the genders for either the pre-intake or post-intake samples (analyzed with *t*-test).

**Figure 5 ijms-26-05706-f005:**
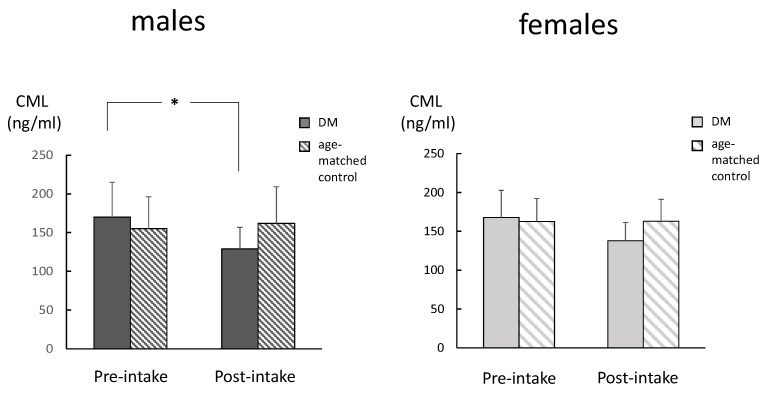
Comparisons between the DM and the age-matched controls for the changes in the CML levels. There were no significant differences in the CML levels, in either the pre-intake or post-intake samples in both genders (analyzed with *t*-test). * *p* < 0.05.

**Figure 6 ijms-26-05706-f006:**
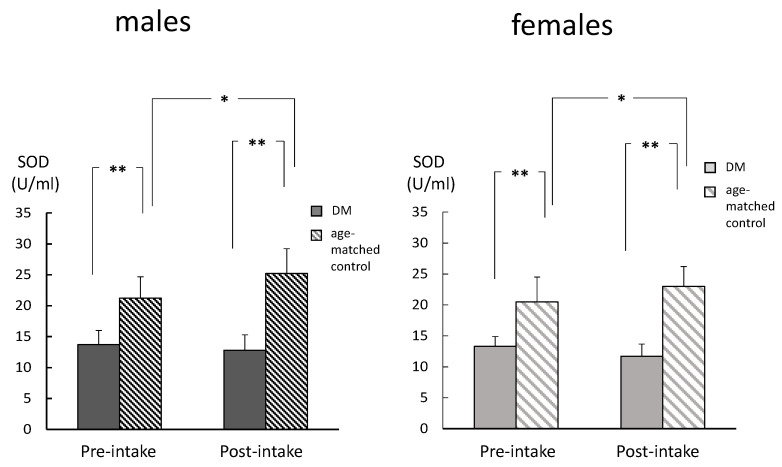
Comparisons between the DM and aged groups for the changes in the SOD levels. The SOD levels were significantly lower in the DM group among both genders, in both the pre-intake and post-intake samples (analyzed with *t*-test). * *p* < 0.05. ** *p* < 0.01.

**Figure 7 ijms-26-05706-f007:**
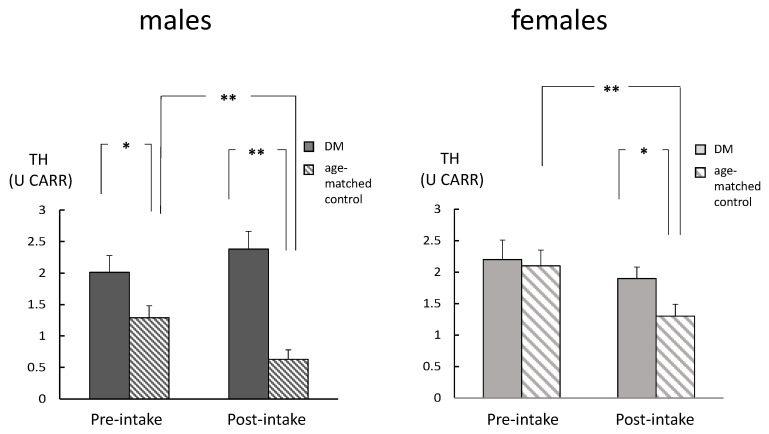
Comparisons between the DM and aged groups for changes in the TH levels. TH levels were significantly higher among males in both the pre-intake and post-intake samples while they were only higher in the post-intake samples among females (analyzed with *t*-test). * *p* < 0.05. ** *p* < 0.01.

**Table 1 ijms-26-05706-t001:** Characteristics of enrolled patients.

		Age-Matched Control *	DM Group	*p*
No.	males	45	10	
	females	55	15	
Age (y/o)	males	74.9 ± 5.5	74.4 ± 5.7	n.s
	females	74.3 ± 5.1	75.1 ± 5.9	n.s
Pre-intake CML (ng/mL)	males	155.2 ± 41.0	163.1 ± 54.9	n.s
	females	162.9 ± 29.4	158.2 ± 34.8	n.s
Post-intake CML (ng/mL)	males	162.1 ± 47.1	134.9 ± 27.7	n.s
	females	163.1 ± 28.0	142.0 ± 23.5	n.s
Pre-intake SOD (U/mL)	males	21.2 ± 3.5	13.7 ± 2.3	<0.0001
	females	20.5 ± 4.0	13.3 ± 1.6	0.0001
Post-intake SOD (U/mL)	males	25.2 ± 4.0	12.8 ± 2.5	<0.0001
	females	23.0 ± 3.2	11.7 ± 2.0	<0.0001
Pre-intake TH (CARR)	males	1.3 ± 0.2	2.0 ± 0.3	0.0345
	females	2.1 ± 0.3	2.2 ± 0.3	n.s
Post-intake TH (CARR)	males	0.6 ± 0.2	2.4 ± 1.3	0.0016
	females	1.3 ± 0.2	1.9 ± 0.2	0.0153

* age-matched control: non-DM patients matched to DM group in age.

**Table 2 ijms-26-05706-t002:** The composition of Ocuvite + Lutein^®^.

Substance	Amount
lutein	6.0 mg
vitamin C	300.0 mg
vitamin E	60.0 mg
vitamin B_2_	3.0 mg
β-carotene	1200.0 μg
niacin	12.0 mg
zinc	9.0 mg
selenium	45.0 μg
copper	0.6 mg
manganese	1.5 mg

## Data Availability

The data that support the findings of this study are available from the corresponding author upon reasonable request.

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
