# Peer review of "The Effects of Lutein-Containing Supplement Intake on Glycation Inhibition Among Diabetic Patients with Cataracts"

_ijms, 2025, doi:10.3390/ijms26125706_

Round 1

Reviewer 1 Report (Previous Reviewer 4)

Comments and Suggestions for Authors

Thank you for the opportunity to review this paper.

Some concerns:

  • lines 107 -110 , authors reported: "In the age-matched controls, there were no significant differences between the CML levels in pre-intake and that in post-intake samples, for either the males or females (Figure 4a). There was a significant correlation between the CML levels in the pre-intake samples and that in the post-intake samples among both genders (Figure 4b)". Could you give a better interpretation of these results in the discussion section? How do you explain this correlation? 
  • lines 196-197: authors reported this sentence "AGEs are presumed to increase when the hyperglycemia progresses due to the further reduction in the capacities in the ROS handling.". The association between AGEs and hyperglycemia is well-established and substantiated by numerous studies (e.g., Brownlee, 2001; Ahmed, 2005), rather than being a speculative assumption. Please edit this sentence adding references like this : https://doi.org/10.1016/j.bone.2023.116884

Author Response

Comment 1: lines 107 -110, authors reported: "In the age-matched controls, there were no significant differences between the CML levels in pre-intake and that in post-intake samples, for either the males or females (Figure 4a). There was a significant correlation between the CML levels in the pre-intake samples and that in the post-intake samples among both genders (Figure 4b)". Could you give a better interpretation of these results in the discussion section? How do you explain this correlation?

Reply: Thank you for your helpful suggestions. We have added further text on the interpretation of these results in our revised manuscript.

Comment 2: lines 196-197: authors reported this sentence "AGEs are presumed to increase when the hyperglycemia progresses due to the further reduction in the capacities in the ROS handling.". The association between AGEs and hyperglycemia is well-established and substantiated by numerous studies (e.g., Brownlee, 2001; Ahmed, 2005), rather than being a speculative assumption. Please edit this sentence adding references like this : https://doi.org/10.1016/j.bone.2023.116884

Reply: In our original text, we tried to say that the levels of AGEs of the DM patients included in the present study were the same as the levels for the non-DM patients because they were well controlled. Thus, if DM control became poor, there would be an increase in the hyperglycemia, and thus, the AGEs would increase in these patients. We apologize for the confusion. As per your comment, we have further revised this text.

Reviewer 2 Report (New Reviewer)

Comments and Suggestions for Authors

The main concern is that much of the discussion should be moved to the introduction.  As it stands, the importance of the research and the knowledge gap is not clear from the introduction.  Meanwhile, the discussion sounds more like a review of their work.  For example, the section containing lines 163 through 178 would provide justification for the present work and should be moved to the introduction.  

The discussion should be focussed just on the interpretation of the current findings. 

Methods- Could the reason for the lack of difference between the DM and the age-matched controls be due to the large variability in the data? Or, that the age-matched controls would have been on a less strict diet and therefore more variable?

Author Response

Comments 1: The main concern is that much of the discussion should be moved to the introduction. As it stands, the importance of the research and the knowledge gap is not clear from the introduction. Meanwhile, the discussion sounds more like a review of their work. For example, the section containing lines 163 through 178 would provide justification for the present work and should be moved to the introduction.

Reply: Thank you for your helpful comments. With regard to the importance of the present study, we have added a sentence to clarify that this is the first investigation on suppressing glycation with antioxidant supplement by measuring AGEs in aqueous humor of the same patient. The descriptions of our previous studies have provided evidence of antioxidative effects of this lutein-containing supplement and background for the current interpretation of the results of our present study. One of our previous works has been described in the Introduction section in order to provide the background for the present study. Another of our previous investigations were mentioned in the Discussion section to describe the possibilities of gender differences. Therefore, we have chosen to retain this information in the Discussion section.

Comments 2: The discussion should be focused just on the interpretation of the current findings.

Reply: In addition to interpretation of the current findings, we believed we should describe the limitations of the current work and suggestions for further potential investigations.

Comments 3: Methods- Could the reason for the lack of difference between the DM and the age-matched controls be due to the large variability in the data? Or, that the age-matched controls would have been on a less strict diet and therefore more variable?

Reply: According to the distributions and our statistical analyses, we believe that it is unlikely that the lack of differences between the two groups was due to the large variability. As we have described in our Discussion section, all of the included patients, which included both DM and non-DM patients, were instructed not to change their diet. Therefore, it is unlikely that the diet would have influenced our results. Furthermore, as the capacities of handling ROS were higher (higher SOD activities and lower TH levels), it is unlikely that diet of the age-matched control was less strict than the DM patients.

Round 2

Reviewer 1 Report (Previous Reviewer 4)

Comments and Suggestions for Authors

The revised paper has been improved, following revisors' suggestions. 

This manuscript is a resubmission of an earlier submission. The following is a list of the peer review reports and author responses from that submission.

Round 1

Reviewer 1 Report

Comments and Suggestions for Authors

The authors of this manuscript evaluated the effect of a supplement rich in antioxidants and lutein on oxidative stress parameters and carboxymethyl-lysine (CML) levels in the vitreous humor of healthy controls and diabetic patients with cataracts in a small clinical trial. The supplement reduced CML levels in male diabetics, but did not reduce hydroperoxide levels or increase superoxide dismutase activity in diabetics but did in healthy subjects. Therefore, the mechanism of action of this supplement is unclear. These and other observations are listed below:

1. In my opinion, the main problem with this study is that the study population was not ideal, as discussed in the limitations of the study in the discussion. It is not clear that the decision to study the cortical cataract was made knowing that the anterior and not the cortical cataract is most affected by diabetes.

2. The other point is that it is difficult to evaluate the effect of an agent with therapeutic potential in people with well-controlled diabetes and without complications. It is possible that no effect was found because it is probable that the medications taken by the patients have their own effect on the parameters measured and that, in the cases where there was a positive effect, the effect was only due to the progression of the pharmacological treatment of diabetes in these patients.

3. Was the effect of lutein supplementation on corneal opacity not evaluated? Oxidative stress does not play the only role in the development of corneal opacity, so it is likely that there was an improvement in cataract independent of the antioxidant effects. For example, the increase in glucose activates the polyol pathway, which leads to an increase in osmolarity due to the excess sorbitol produced. Is it possible that the formulation used improves the osmolarity of the vitreous?

4. The effects of this supplement are attributed to its lutein content. How can it be excluded that the other antioxidant compounds in this formulation are responsible for the observed effects when some of them, such as beta-carotene and vitamin C, are present in concentrations several orders of magnitude higher than lutein?

5. How is it possible that there are significant differences between the groups compared when the dispersion of the data is so great that the error bars between the different groups overlap?

6. Superoxide anion is the major ROS produced when there is mitochondrial dysfunction due to activation of NADPH oxidases, and it is the major species that mediates damage when there is accumulation of AGEs. However, the superoxide anion does not produce hydroperoxides (i.e., it is not a ROS capable of initiating lipid peroxidation). What was the rationale for measuring hydroperoxides if the superoxide radical is not a mediator of hydroperoxide formation? Furthermore, if hydroperoxides were measured, why not measure the levels or activity of glutathione peroxidase (GPX), which, unlike SOD, are enzymes involved in the detoxification of hydroperoxides?

Author Response

The authors of this manuscript evaluated the effect of a supplement rich in antioxidants and lutein on oxidative stress parameters and carboxymethyl-lysine (CML) levels in the vitreous humor of healthy controls and diabetic patients with cataracts in a small clinical trial. The supplement reduced CML levels in male diabetics, but did not reduce hydroperoxide levels or increase superoxide dismutase activity in diabetics but did in healthy subjects. Therefore, the mechanism of action of this supplement is unclear. These and other observations are listed below:

Response: We thank you for your helpful suggestions. Also, we would like to clarify that we evaluated the aqueous humor and not the vitreous in this study. We also apologize for any confusion that might have resulted from the presentation of our text in the previous version of the manuscript.

Comment 1: In my opinion, the main problem with this study is that the study population was not ideal, as discussed in the limitations of the study in the discussion. It is not clear that the decision to study the cortical cataract was made knowing that the anterior and not the cortical cataract is most affected by diabetes.

Response: Cortical cataracts were indeed included in this study, as these make up the majority of cataracts that are seen in patients in Japan. Furthermore, this study included DM patients without DM complications, as the severities of the DM complications could potentially lead to confusion regarding our evaluations for the readers of this manuscript.

Comment 2:  The other point is that it is difficult to evaluate the effect of an agent with therapeutic potential in people with well-controlled diabetes and without complications. It is possible that no effect was found because it is probable that the medications taken by the patients have their own effect on the parameters measured and that, in the cases where there was a positive effect, the effect was only due to the progression of the pharmacological treatment of diabetes in these patients.

Response: All patients in the DM group had been treated for DM prior to the start of our evaluation. Thus, the effects of the antidiabetic medication should be equal between the pre- and post-intake samples.

Comment 3: Was the effect of lutein supplementation on corneal opacity not evaluated? Oxidative stress does not play the only role in the development of corneal opacity, so it is likely that there was an improvement in cataract independent of the antioxidant effects. For example, the increase in glucose activates the polyol pathway, which leads to an increase in osmolarity due to the excess sorbitol produced. Is it possible that the formulation used improves the osmolarity of the vitreous?

Response: We agree that oxidative stress does not play the only role in corneal opacities. To the best of our knowledge, corneal opacities occur in different ways from that for lens opacities. Also, nutrients are supplied to the cornea through tear and limbal vessels. In contrast, they are supplied to the lens through the aqueous humor. Thus, this is the reason why the effects on the corneal opacities were not evaluated. Moreover, this is also the reason why we chose the aqueous humor for investigation. The vitreous, including the osmolarity, were not evaluated in this present study, as the supplement evaluated in this study does not decrease sugar levels. Therefore, if there was a potential improvement in osmolarity of the vitreous, the present study was not set up to consider this point.

Comment 4: The effects of this supplement are attributed to its lutein content. How can it be excluded that the other antioxidant compounds in this formulation are responsible for the observed effects when some of them, such as beta-carotene and vitamin C, are present in concentrations several orders of magnitude higher than lutein?

Response: We agree with your opinion. It is possible that the effects were not only due to lutein but could also potentially be related to the synergic effects of the antioxidants included in this supplement. We apologize for the confusion regarding this issue. We have added an additional paragraph to our discussion section to clarify this issue.

Comment 5: How is it possible that there are significant differences between the groups compared when the dispersion of the data is so great that the error bars between the different groups overlap?

Response: All the statistical analyses were performed using SPSS. As the amount of collectable aqueous humor was small and the concentrations of substances in the aqueous humor were low, this meant there could be large variations among the examined individuals. Also, even though error bars overlapped, as shown in Figure 1a, CML distribution was lower in post-intake samples than that in pre-intake samples. Also, as shown in Figure 1b, all the CML level values for males are located on the right portion of the graph below the regression line, which indicates decreases in the CML levels after the intake. Combining the results of analyses with SPSS, we were able to conclude that there were indeed significant decreases in CML among male DM patients.

Comment 6: Superoxide anion is the major ROS produced when there is mitochondrial dysfunction due to activation of NADPH oxidases, and it is the major species that mediates damage when there is accumulation of AGEs. However, the superoxide anion does not produce hydroperoxides (i.e., it is not a ROS capable of initiating lipid peroxidation). What was the rationale for measuring hydroperoxides if the superoxide radical is not a mediator of hydroperoxide formation? Furthermore, if hydroperoxides were measured, why not measure the levels or activity of glutathione peroxidase (GPX), which, unlike SOD, are enzymes involved in the detoxification of hydroperoxides?

Response: We agree that glutathione peroxidase is one of indicators of the detoxication of hydroperoxides. However, as only a small amount of aqueous humor can be collected, this means that the number of measurement items is limited. Thus, in the present study, we measured superoxide dismutase activities (SOD) as an indicator of scavengers of ROS and total hydroperoxides (TH, including hydrogen peroxide and the peroxides of lipids, peptides, proteins, nucleic acids and nucleotides) in order to determine the status of the peroxidation. The levels of the total hydroperoxide might also represent the activities of the enzymes that scavenge hydroperoxides.

Reviewer 2 Report

Comments and Suggestions for Authors

The manuscript reports the effects of the intake of Ocuvite + Lutein® (a product containing a mixture of compounds with antioxidant proprieties) on glycation and oxidation processes in diabetic patients and comes as a continuation of the authors’ previous work on the above-mentioned product. The study is well conducted, and the manuscript is likely to be of interest to scientists working in the management of age-related eye diseases and conditions.

The manuscript can be considered for publication in the International Journal of Molecular Sciences with minor revisions with respect to a disambiguation issue. The authors are kindly advised to consider stressing the fact that the antioxidant effect was proven for a product that is a mixture of several compounds endowed with antioxidant properties. The study does not show direct experimental proof that lutein is the ingredient in the formulation that plays the key role in reducing glycation and increasing the antioxidative defense system in the eye; it appears however that, throughout the manuscript, the authors stress the importance of the lutein in the formulation instead of presenting the fact that the observed clinical effects are due to the combined effects of the multicomponent formulation (please note that even in the keywords section, lutein is one of the keywords while all the other components with antioxidant activity are omitted). As such the conclusions of the study may be misleading.

Also, the composition of Ocuvite + Lutein® is shown in Table 2, not Table 1. Please check line 228

Author Response

Comment 1: The manuscript can be considered for publication in the International Journal of Molecular Sciences with minor revisions with respect to a disambiguation issue. The authors are kindly advised to consider stressing the fact that the antioxidant effect was proven for a product that is a mixture of several compounds endowed with antioxidant properties. The study does not show direct experimental proof that lutein is the ingredient in the formulation that plays the key role in reducing glycation and increasing the antioxidative defense system in the eye; it appears however that, throughout the manuscript, the authors stress the importance of the lutein in the formulation instead of presenting the fact that the observed clinical effects are due to the combined effects of the multicomponent formulation (please note that even in the keywords section, lutein is one of the keywords while all the other components with antioxidant activity are omitted). As such the conclusions of the study may be misleading.

Response: Thank you for your helpful comments and we agree with your observations. We also considered that the anti-oxidative effects could potentially be from not only lutein but also could be due to synergic effects from the other combined antioxidants found in this supplement. We apologize for any confusion that might have arisen from our unclear description of the components evaluated. To avoid any misunderstanding on the part of the reader, we have added a paragraph to our discussion section to further clarify this issue.

Comment 2: Also, the composition of Ocuvite + Lutein® is shown in Table 2, not Table 1. Please check line 228

Response: We apologize for the mistake in our Table numbering. This has now been corrected.

Reviewer 3 Report

Comments and Suggestions for Authors

The authors presented a manuscript on the effects of a commercial compound (Ocuvite-Lutein) on total hydroperoxide (TH) levels, Carboxymethyl-lysine (CML, an advanced glycation end-product or AGE), and superoxide dismutase (SOD) activity in the aqueous humor of patients with binocular cataracts. Measurements were taken first in one eye (pre-intake) and subsequently in the other eye (post-intake). The enrolled patients were divided into two experimental groups: one consisting of diabetic patients (DM) and the other of age-matched non-diabetic controls.

The data indicate that CML levels decreased after intake in the DM group, while no changes were observed in the control group. SOD activity was significantly lower, and TH levels significantly higher, in the DM group compared to controls, both before and after intake.

The data provided by the authors are largely observational and do not elucidate molecular mechanisms underlying the antioxidant effects of the nutraceuticals consumed by the patients. Moreover, the antioxidant properties of individual compounds in the pharmaceutical product on AGEs and SOD activity have already been described in numerous studies. These criticisms are well summarized in the "Conclusion" paragraph, where a sentence with a generic and substantially inconclusive meaning is provided.

The authors should evaluate the effects of Ocuvite-Lutein on clinical parameters related to potential post-operative complications of cataract surgery or on cataract incidence before surgery.

Another critique is that lutein is often emphasized in the manuscript as if it were the sole antioxidant substance, whereas the pharmaceutical product (Ocuvite-Lutein) contains several compounds, all with antioxidant properties.

Author Response

Comment 1: The authors should evaluate the effects of Ocuvite-Lutein on clinical parameters related to potential post-operative complications of cataract surgery or on cataract incidence before surgery.

Response: Thank you for your helpful comments. The most common post-operative complication is inflammation. However, due to improvements in the recent surgical technique, this type of inflammation is now usually very mild and thus, can be difficult to use for the purpose of evaluating the effects of any antioxidants. Concerning the incidence of cataracts, several investigations have reported finding an association between decreases in the cataract incidence and antioxidants, including lutein (Moeller, 2008; Christen; 2008). These references have been additionally cited in the main text line 47 as references No.27 and 28.

Comment 2: Another critique is that lutein is often emphasized in the manuscript as if it were the sole antioxidant substance, whereas the pharmaceutical product (Ocuvite-Lutein) contains several compounds, all with antioxidant properties.

Response: We agree with the above point. We also considered that the anti-oxidative effects were not only from lutein but that synergic effects were also possible from the other combined antioxidants. We apologize for any confusion associated with our original text. We have added a further paragraph that helps to clarify this issue in our revised Discussion section.

Reviewer 4 Report

Comments and Suggestions for Authors

Thank you for the opportunity to review this paper. Some concerns:

Introduction: Line 47: "Many reviews have investigated nutrients and their effect on eye health." Please add references for this sentence

Results: Please unify and improve the lines 60-66 with lines 79-86. The sentences of these lines refer to the same content. 

Conclusions: lines 313-314: "These results trongly suggest the possibilities of inhibiting cataract progression and diabetic complications in this patient group". This statement is too categorical. The study did not assess cataract progression in patients but only the change in ROS levels before and after the intake of antioxidants with lutein. To claim that lutein supplements reduce the onset of cataracts, longitudinal studies with long follow-ups would be necessary. Please revise the conclusions accordingly. The same should be done for the abstract

Author Response

Comment 1: Introduction: Line 47: "Many reviews have investigated nutrients and their effect on eye health." Please add references for this sentence

Response: Thank you for your helpful comments. We have added references for this statement.

Comment 2: Results: Please unify and improve the lines 60-66 with lines 79-86. The sentences of these lines refer to the same content.

Response: Lines 60-66 are a description of the results in our main text while lines 79-86 refer to information for the legend of Figure 1. Thus, we have combined the text for line 79 with the text for the title of Figure 1. We apologize for any confusion associated with our original text.

Round 2

Reviewer 1 Report

Comments and Suggestions for Authors
  1. The argument given about the validity of the statistical analysis of the data is not convincing.
  2. On the other hand, assigning a role to lutein in the observed effects is confusing because the formulation used contains a wide variety of antioxidants with therapeutic potential. Therefore, the mention in the title of the paper that lutein is the component with the anti-glycation effect is not entirely correct.
  3. Likewise, the argument given to justify measuring SOD activity  and not GPX is not convincing. Nor can a mechanism of action for this formulation be discerned from the data presented.

Author Response

Comment 1: The argument given about the validity of the statistical analysis of the data is not convincing.

Response: We understand the reviewer’s concern and have consulted a statistics expert. Combining the scatter plot graph and the results of analyses with IBM®SPSS Statistics 28, we believe the original statistical approach was valid.

Comment 2: On the other hand, assigning a role to lutein in the observed effects is confusing because the formulation used contains a wide variety of antioxidants with therapeutic potential. Therefore, the mention in the title of the paper that lutein is the component with the anti-glycation effect is not entirely correct.

Response: As we described our previous response, we agree that not only lutein but all the antioxidants included in this supplement likely play roles in anti-glycation and anti-oxidation. A paragraph addressing this has been added to the Discussion section. Since it was deemed inappropriate to specify the product in the title using the brand name that emphasizes the lutein content (Ocuvite + Lutein®), we referred to it is as a lutein-containing antioxidant supplement to distinguish it from other supplements, and also to convey the nuance that the supplement is formulated with lutein. We also chose to use "lutein-containing antioxidant supplement" as it is similar to wording used in titles of our previously published work.

Comment 3: Likewise, the argument given to justify measuring SOD activity and not GPX is not convincing. Nor can a mechanism of action for this formulation be discerned from the data presented.

Response: We agree that Gpx is an important scavenger of hydrogen peroxide, but not the only one. As we described in our previous response, only a small amount of aqueous humor can be sampled, which limited the number of measurements that could be conducted. To the best of our knowledge, superoxide plays a role in peroxidation and the production of hydroperoxides (including hydrogen peroxide and the peroxides of lipids, peptides, proteins, nucleic acids and nucleotides), and superoxide anions have been reported to be an initiator of microsomal lipid peroxidation in several investigations (such as in Afanas'ev IB, et al. Biochem Biophys. 1993;302(1):200-5.). Therefore, we measured the scavenging activities of superoxide in present study and our previously published works. Also, as described in our previous response, total hydroperoxide levels might also reflect the activities of all the enzymes that scavenge hydrogen peroxides, including Gpx.

As we described in the Discussion section, especially in the second paragraph, oxidation and glycation are tightly link. The results of our study revealed that CML decreased after intake of the lutein-containing antioxidant supplement, suggesting this supplement may inhibit glycation.

Reviewer 3 Report

Comments and Suggestions for Authors

As previously noted in the initial review, the manuscript fails to present research advances regarding Ocuvite-Lutein's effects on cataract surgery that meet the scientific rigor expected by IJMS.

Author Response

Comment: As previously noted in the initial review, the manuscript fails to present research advances regarding Ocuvite-Lutein's effects on cataract surgery that meet the scientific rigor expected by IJMS.

Response: We apologize for the confusion. Our focus was not on the effects of the lutein-containing antioxidant supplement in preventing complications or other factors related to cataract surgery. The purpose of our study was to determine oxidation and glycation status in aqueous humor in diabetic patients, and aqueous humor can be only collected during cataract surgery. This is why patients undergoing cataract surgery were included in the study.

Round 3

Reviewer 1 Report

Comments and Suggestions for Authors

None of my concerns have been convincingly addressed. Not a statistical or mathematical response to my concerns regarding to statistical analysis has given.